# Masked Image Residual Learning for Scaling Deeper Vision Transformers

**Guoxi Huang**
Baidu Inc.
huangguoxi@baidu.com

**Hongtao Fu**
Huazhong University of Science and Technology
m202173233@hust.edu.cn

**Adrian G. Bors**[†]
University of York
adrian.bors@york.ac.uk

## Abstract

Deeper Vision Transformers (ViTs) are more challenging to train. We expose a degradation problem in deeper layers of ViT when using masked image modeling (MIM) for pre-training. To ease the training of deeper ViTs, we introduce a self-supervised learning framework called **M**asked **I**mage **R**esidual **L**earning (**MIRL**), which significantly alleviates the degradation problem, making scaling ViT along depth a promising direction for performance upgrade. We reformulate the pre-training objective for deeper layers of ViT as learning to recover the residual of the masked image. We provide extensive empirical evidence showing that deeper ViTs can be effectively optimized using MIRL and easily gain accuracy from increased depth. With the same level of computational complexity as ViT-Base and ViT-Large, we instantiate $4.5\times$ and $2\times$ deeper ViTs, dubbed ViT-S-54 and ViT-B-48. The deeper ViT-S-54, costing $3\times$ less than ViT-Large, achieves performance on par with ViT-Large. ViT-B-48 achieves 86.2% top-1 accuracy on ImageNet. On one hand, deeper ViTs pre-trained with MIRL exhibit excellent generalization capabilities on downstream tasks, such as object detection and semantic segmentation. On the other hand, MIRL demonstrates high pre-training efficiency. With less pre-training time, MIRL yields competitive performance compared to other approaches. Code and pretrained models are available at: *https://github.com/russellllaputa/MIRL*.

## 1 Introduction

Transformer architecture [46] has become the de-facto standard in natural language processing (NLP). A major driving force behind the success of Transformers in NLP is the self-supervised learning method called masked language modeling (MLM) [10]. MLM significantly expends the generalization capabilities of Transformers, with the underlying principle being very intuitive - removing portions of a sentence and learning to predict the removed content. Recent advancements in computer vision have been profoundly inspired by the scaling successes of Transformers in conjunction with MLM in NLP, successively introducing the Vision Transformer (ViT) [13] and masked image modeling (MIM) for training generalizable vision models. The concept of MIM is as straightforward as MLM; its pre-training objective is to predict masked image patches based on the unmasked image patches, thereby capturing rich contextual information.

This paper first reveals that MIM can induce negative optimization in deeper layers of ViT, which not only constrains the generalization performance of ViT but also hinders its scaling along the depth dimension. Previous work [55, 37] suggests that deeper layers of ViT are more properly pre-trained by using MIM, the conclusions of which contradict our observation. Another branch of work [6, 50, 4, 25, 37] tentatively suggests that, due to the lack of semantic information in MIM, the shallower layers of ViTs are more effectively pre-trained than the deeper layers. In our preliminary experiments in Sec. 2.2, we demonstrate that replacing the deeper Transformer blocks pre-trained by using MIM with randomly initialized blocks does not degrade performance, which supports our

---

[†]Corresponding author. Work done when H. Fu was an intern at Baidu.

37th Conference on Neural Information Processing Systems (NeurIPS 2023).

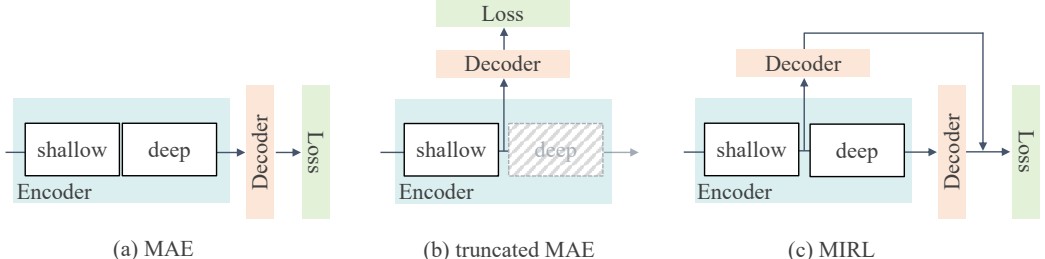

Figure 1: Three MIM pre-training schemes. For simplicity, the diagram omits the random masking process. A complete ViT consists of shallow and deep parts. The dashed box indicates the part of the model that is not involved in the pre-training process.

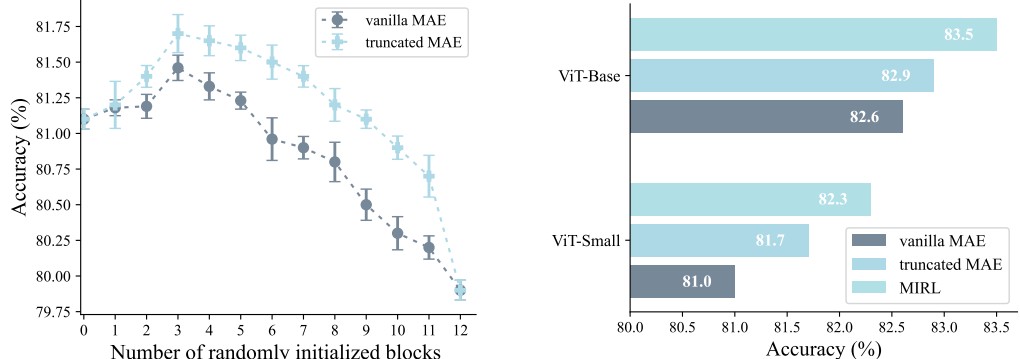

Figure 2: Truncated MAE vs. MAE. The $x$-axis represents the number of blocks replaced with randomly initialized blocks from the encoder's end after pre-training. ViT-S is used as the encoder to better observe the differences.

Figure 3: Comparison of MIRL, truncated MAE, and MAE. To maintain the same decoding computational cost, each decoder in the MIRL model contains 2 blocks, while other models have 4 decoder blocks.

statement. We hypothesize the negative pre-training effect enforced on the deeper layers is due to depth restriction, which can be regarded as a degradation problem occurring in deeper ViTs.

We address the pre-training degradation problem in deeper layers of ViT by introducing a **M**asked **I**mage **R**esidual **L**earning (**MIRL**) framework. We establish a multi-decoding process by segmenting the encoding blocks according to their depth levels. Instead of letting the entire autoencoder learn to reconstruct the masked content, MIRL encourages the deep layers to learn latent features that are beneficial for recovering the image residual, distinct from the main image component. The diagram of the MIRL with 2 segments is illustrated in Figure 1c, where we divide the encoder into shallow and deep segments, and append a separate decoder to each. The shallow segment learns to reconstruct the main component of the masked content, while the deep segment is explicitly reformulated as learning the image residual. The MIRL framework is essentially equivalent to shifting the pre-training objective of deeper layers of ViT from image reconstruction to image residual reconstruction. This simple yet effective concept of image residual reconstruction significantly alleviates the degradation problem in deeper layers of ViT, making scaling ViTs along depth a promising direction for improving performance. By extending the MIRL framework and dividing the encoder into more segments, as illustrated in Figure 4, we can train deeper ViTs and readily achieve accuracy gains from substantially increased depth. Consequently, we instantiate deeper encoders: ViT-B-24, ViT-B-48, and ViT-S-54, comprising 24, 48, and 54 Transformer blocks, respectively. Notably, with similar computational complexity, our deeper ViT variants deliver considerably better generalization performance than the wider ViT encoders (*e.g.* ViT-S-54 *vs.* ViT-B, ViT-B-48 *vs.* ViT-L), thanks to the increased depth. Meanwhile, our experiments in Sec. 4.2 demonstrate that employing additional feature-level objectives [12, 6] or VGG loss [26, 11] can further improve performance, suggesting that the improvement directions of feature-level loss and MIRL are orthogonal and can complement each other.

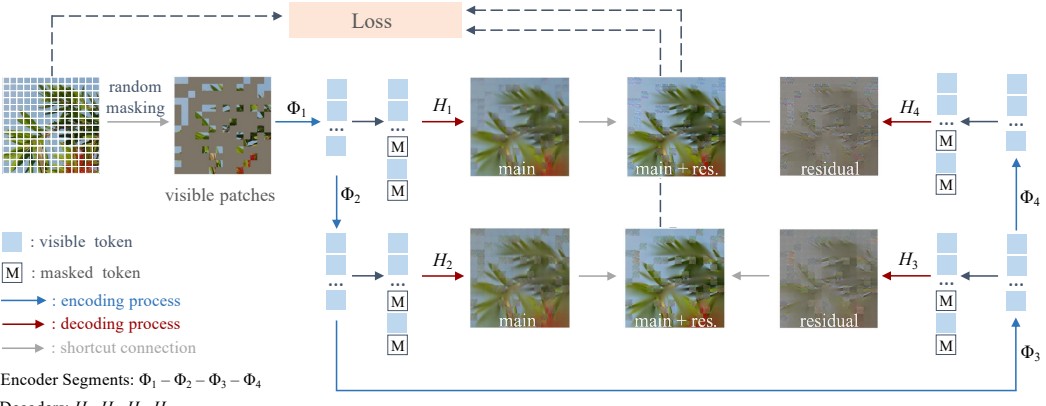

Figure 4: Example of the MIRL framework. The Transformer blocks in the ViT encoder are split into four segments, $\Phi_1$, $\Phi_2$, $\Phi_3$, and $\Phi_4$. The output of each segment is then sent to its corresponding decoder. Shortcut connections are established between the shallower decoders $H1$, $H_2$ and deeper decoders $H4$, $H_3$, enabling deeper decoders to predict the masked image residual.

## 2 Preliminaries

### 2.1 Revisit masked image modeling

Following the paradigm of Masked AutoEncoder(MAE) [21], the input image is split into a set of non-overlapping patches $\mathbf{x} = \{\mathbf{x}^i\}_{i=1}^N$, where $\mathbf{x}^i \in \mathbb{R}^{P^2 C}$ denotes the image patch in the $i$-th position with $P^2$ resolution and $C$ channels. The image patches are further tokenized into visual tokens via a linear projection, denoted by $\mathbf{z}_0 = \{\mathbf{z}_0^i\}_{i=1}^N$ [^2]. Subsequently, a random sampling strategy divides the indexes of patches into two subsets, $\mathcal{V}$ and $\mathcal{M}$, where $\mathcal{V} \cap \mathcal{M} = \emptyset$. The image patches ($\mathbf{x}^{\mathcal{V}}$) and visual tokens ($\mathbf{z}_0^{\mathcal{V}}$) with indexes in $\mathcal{V}$ are considered to be visible to the encoder. Thereafter, the encoder with $L$ blocks only takes as input visible tokens $\mathbf{z}_0^{\mathcal{V}}$, and maps them to embedding features $\mathbf{z}_L^V$. The objective of MIM is to predict the unseen content from $\mathbf{x}^{\mathcal{M}}$ by employing a decoder. A learnable mask token $\mathbf{e}_{[\mathrm{M}]}$ is introduced after the encoder, which is placed in $\mathcal{M}$ masked positions. Then the full set of encoded patches and mask tokens is processed by a small decoder to reconstruct the original image in pixels. The architecture of MAE can be described as:

$$\mathbf{z}_\ell^{\mathcal{V}} = F_\ell(\mathbf{z}_{\ell-1}^{\mathcal{V}}), \qquad \ell = 1...L \tag{1}$$

$$\mathbf{u} = \mathbf{z}_L^{\mathcal{V}} \cup \{\mathbf{e}_{[\mathrm{M}]} : i \in \mathcal{M}\}_{i=1}^N, \tag{2}$$

$$\hat{\mathbf{x}} = H(\mathbf{u}), \tag{3}$$

where $F_\ell(\cdot)$ refers to the $\ell$-th Transformer block in the encoder, $H(\cdot)$ denotes a shallow decoder. The objective loss of MIM is given by

$$\mathcal{L}^{\mathrm{pixel}} = \frac{1}{|\mathcal{M}|} \sum_{i \in \mathcal{M}} \frac{1}{P^2 C} \left\| \hat{\mathbf{x}}^i - \mathbf{x}^i \right\|_2^2, \tag{4}$$

where the reconstruction loss is only calculated in the masked positions.

### 2.2 A deep dive into autoencoders for MIM

We present three distinct autoencoder architectures in Figure 1 to clarify our motivation. Figure 1a shows the MAE framework [21]. Figure 1b depicts a truncated MAE with only the shallow ViT blocks in the encoder. Figure 1c presents a simplified MIRL with two decoders connected to the shallow and deep encoder segments. Models are pre-trained for 300 epochs with the same hyperparameters. Full setup details are in Appendix A of the supplementary materials.

[^2]: We still perform the position embedding addition and class token concatenation processes as in ViT, but these steps are omitted for notational simplicity.

**Observation I:** *MIM pre-training can induce negative optimization in deeper layers of ViT.* However, due to the overwhelmingly positive pre-training effect that MIM bestows upon the earlier blocks, its adverse influence on the latter blocks remains undiscovered. Given a ViT encoder pre-trained with MIM (Figure 1a), we substitute the pre-trained weights in the latter blocks of the encoder with the random parameters and subsequently fine-tune the encoder on ImageNet-1K. The curve plot of vanilla MAE, depicted in Figure 2, illustrates that applying random re-initialization to deeper blocks from the end of the encoder can improve performance. The initial point in Figure 2 indicates the result of an MAE model that has been pre-trained and subsequently fine-tuned without any random initialization. As more shallow blocks are randomly re-initialized, accuracy declines. Intriguingly, random initialization generalizes better than MIM pre-training in deeper layers of ViT, defying intuition.

**Observation II:** *Performing MIM pre-training on fewer layers can lead to better efficiency and effectiveness.* A truncated MAE illustrated in Figure 1b requires less pre-training time than the vanilla MAE while still achieving better or comparable performance. For fine-tuning, we initialize a complete ViT model with pre-trained weights from the truncated MAE. Regarding the blocks that are not included in the truncated MAE, we apply random initialization. As shown in Figure 2, when truncating 3 blocks from the end of the encoder, the fine-tuned model has better accuracy than the rest ones. By using the truncated MAE, we only pre-train 4 blocks and achieve similar fine-tuning accuracy as the vanilla MAE, reducing pre-training cost by 66%.

**Observation III:** *Learning to recover image residual is a more productive pre-training objective.* A simplified version of MIRL, shown in Figure 1c, formulates the pre-training objective for deeper layers as learning image residual, promoting more vivid image detail recovery and imposing a positive pre-training effect on deeper layers. Figure 3 demonstrates that MIRL achieves the highest fine-tuning accuracy among the three MIM pre-training schemes.

In summary, Observations I and II expose a pre-training degradation problem in ViT's deeper layers, leading to sub-optimal solutions. The same issue is also observed in the BEiT [1] paradigm, potentially attributed to depth limitations in MIM. In Observation III, we introduce MIRL to alleviate the degradation problem in deeper layers. The rest of the paper demonstrates how we employ MIRL to tackle the challenges of training deeper Vision Transformers.

## 3    Method

### 3.1    Masked Image Residual Learning (MIRL)

Upon observing that deeper layers pre-trained by MIM underperform against those with random initialization, we infer that the weight parameters of these deeper layers have indeed been updated during MIM pre-training, but in an unfavorable direction. In contrast, the shallower layers demonstrate improved performance after MIM pre-training. This leads us to speculate that the depth of the layers could be the root cause of the degradation problem.

To alleviate degradation in deeper ViTs during pre-training, we propose letting the deeper Transformer blocks learn to predict the residual of the masked image, rather than directly predicting the masked image itself. An overview of the MIRL framework is illustrated in Figure 4. Specifically, the encoder is partitioned into multiple segments, with each segment being followed by a separate small decoder. Subsequently, we establish shortcut connections between the shallower and deeper decoders. We underscore that these shortcut connections constitute the core of our method. This configuration fosters a seamless collaboration between very shallow and deep Transformer blocks in corrupted image reconstruction: the shallower segment learns to reconstruct the main component of the masked image, while the deeper segment learns the image residual. During pre-training, the established shortcut connections enable back-propagation to affect both the deeper and shallower layers simultaneously. This intertwined relationship between the deeper and shallower layers implies that the pre-training should either guide both towards a beneficial direction or lead both astray. With the introduction of the MIRL framework, our experimental results indicate that the shallower layers have, in essence, steered the deeper layers towards a more favorable direction.

Formally, we reformulate the encoder in Eq. (1) by evenly grouping the encoding Transformer blocks into $G$ segments:

$$\mathbf{z}_g^{\mathcal{V}} = \Phi_g(\mathbf{z}_{g-1}^{\mathcal{V}}), \qquad g = 1...G, \tag{5}$$

where $\Phi_g$ denotes a stack of encoding blocks in the $g$-th segment. In the output of each encoding segment $\mathbf{z}_g^{\mathcal{V}}$, the masked positions are filled with a shared masked token $\mathbf{e}_{[\mathrm{M}]}$, denoted as $\mathbf{u}_g = \mathbf{z}_g^{\mathcal{V}} \cup \{\mathbf{e}_{[\mathrm{M}]} : i \in \mathcal{M}\}_{i=1}^N$. Subsequently, for the $g$-th segment and the $(G-g+1)$-th segment selected from bottom-up and top-down directions, two separate decoders $H_g$ and $H_{G-g+1}$ are appended for feature decoding. Let us consider that the $g$-th shallower segment learns a mapping function $\hat{\mathbf{x}}_g = H_g(\mathbf{u}_g)$ producing the main component of the reconstructed image $\hat{\mathbf{x}}_g$. Thereafter, we let $\hat{\xi}_g = H_{G-g+1}(\mathbf{u}_{G-g+1})$ from the $(G-g+1)$-th deeper segment asymptotically approximate the residual $\xi_g = \mathbf{x} - \hat{\mathbf{x}}_g$. The objective loss $\mathcal{L}_g$ for the $g$-th segment is defined as:

$$\mathcal{L}_g = \frac{1}{|\mathcal{M}|} \sum_{i \in \mathcal{M}} \frac{1}{P^2 C} \|\xi_g^i - \hat{\xi}_g^i\|_2^2 = \frac{1}{|\mathcal{M}|} \sum_{i \in \mathcal{M}} \frac{1}{P^2 C} \|\mathbf{x}^i - \hat{\mathbf{x}}_g^i - \hat{\xi}_g^i\|_2^2. \tag{6}$$

Different from the residual learning in [23], our image residual learning would not fit an identity mapping, considering that the inputs to the two segments are different. See Appendix B.2 for further discussions on an alternative form of $\mathcal{L}_g$. The final loss function is formed by accumulating all $\frac{2}{G}$ reconstruction loss terms:

$$\mathcal{L}_{\text{total}} = \sum_{g=1}^{\frac{G}{2}} \lambda_g \mathcal{L}_g, \tag{7}$$

where $\lambda_g$ is the scaling coefficient, which is set to $\frac{2}{G}$ by default. Additional pre-training objective losses, such as the feature-level loss used in [6, 54] and the VGG loss [26], can be employed to enhance performance. We provide the definitions of other loss terms in Appendix B.1. However, as indicated in the ablation study in Sec.4.2, incorporating additional loss terms introduces non-negligible overhead during pre-training. By default, we solely utilize the per-pixel loss defined in Eq.(6).

**Densely Interconnected Decoding (DID).** We design a densely interconnected decoding (DID) module, inserted into the decoders across different segments, enabling access to the features produced by previous segments. DID allows subsequent segments to avoid relearning features already acquired in earlier segments, thereby enhancing representation diversity. Immediately following the self-attention module in the first Transformer block in decoder $H_g$, we insert a DID module, which is ingeniously implemented using the Multi-Head Attention mechanism, $\mathrm{MHA}(Q, K, V)$:

$$\mathrm{DID}(\mathbf{u}_g, \mathbf{z}_{g-1}^{\mathcal{V}}, ..., \mathbf{z}_0^{\mathcal{V}}) = \mathrm{MHA}(\mathbf{u}_g, [\mathbf{z}_{g-1}^{\mathcal{V}}, ..., \mathbf{z}_0^{\mathcal{V}}], [\mathbf{z}_{g-1}^{\mathcal{V}}, ..., \mathbf{z}_0^{\mathcal{V}}]), \tag{8}$$

where the query $Q$ is $\mathbf{u}_g$, while the key and value $K, V$ are $[\mathbf{z}_{g-1}^{\mathcal{V}}, ..., \mathbf{z}_0^{\mathcal{V}}]$. The dense interconnection property of the DID module is rendered by the concatenation form in MHA's key $K$ and value $V$ where we concatenate the output from previous encoding segments, thereby enabling feature integration at different depth levels.

## 3.2 Scaling to deeper ViT

Without modifying the Transformer block design, we instantiate deeper ViT variants by simply stacking more blocks. We consider the embedding hidden sizes in the Transformer block of 384 and 768. The details of the deeper ViTs are provided in Table 1. Choosing an appropriate hidden size is non-trivial for scaling up Vision Transformers, considering that a larger hidden size (*e.g.*, 1024 adopted in ViT-Large and 1280 adopted in ViT-Huge) can cause instability due to very large values in attention logits, leading to (almost one-hot) attention weights with near-zero entropy. The instability of very wide ViTs is also reported in [21, 9].

Table 1: Details of Vision Transformer scaling along the depth dimension.

| Model | Depth | Hidden size | MLP size | Heads |
|---|---|---|---|---|
| ViT-S-54 | 54 | 384 | 1536 | 12 |
| ViT-B-24 | 24 | 768 | 3072 | 12 |
| ViT-B-48 | 48 | 768 | 3072 | 12 |

By utilizing MIRL, we demonstrate that deeper ViTs exhibit stronger or comparative generalization capabilities when compared to their shallower and wider counterparts. With similar computational

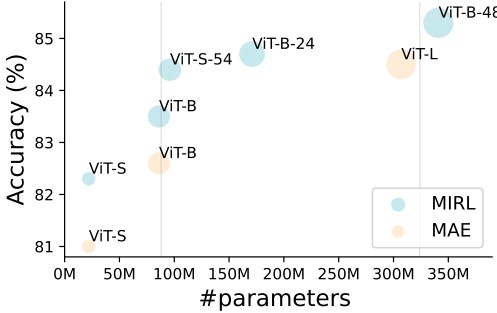 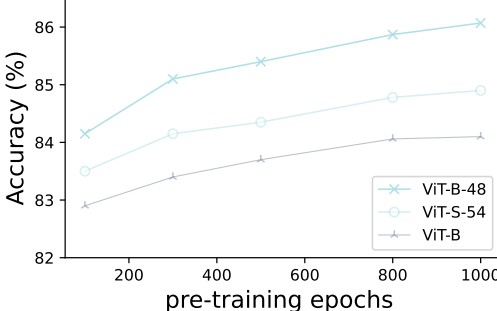

Figure 5: Fine-tuning evaluation on ImageNet versus model size. With a similar complexity, deeper ViTs outperform shallower ViTs. The models are pre-trained for 300 epochs.

Figure 6: Training schedules. The performance of deeper ViTs has not yet plateaued even after 1000 pre-training epochs. We employ a step decay learning rate scheduler.

complexity, ViT-S-54 generalizes better than ViT-B. With only 31% of the computational cost of ViT-L, ViT-S-54 delivers performance on par with ViT-L. Notably, ViT-B-48 not only that it achieves higher performance but also provides a more stable training than ViT-L. This suggests that deepening ViTs could be a promising direction for enhancing vision model performance. Furthermore, MIRL helps to alleviate the training difficulties typically encountered in deeper ViTs, unlocking their potential and making them more effective for a variety of downstream tasks.

# 4   Experiment

The proposed MIRL method is evaluated on image classification, object detection and semantic segmentation tasks. All models are pre-trained on ImageNet-1K and then fine-tuned in downstream tasks. The input size is $224 \times 224$, which is split into 196 patches with a size of $16 \times 16$.

**Pre-training setup.** We pre-train all models on the training set of ImageNet-1K with 32 GPUs. By default, ViT-B-24 is divided into 4 segments, while ViT-S-54 and ViT-B-48 are split into 6 segments, and others into 2. Each appended decoder has 2 Transformer blocks with an injected DID module. We follow the setup in [21], masking 75% of visual tokens and applying basic data augmentation, including random horizontal flipping and random resized cropping. Full implementation details are in Appendix A.

## 4.1   Instantiations of deeper ViT

We compare the performance of the deeper ViTs detailed in Sec.3.2 with the ViT instantiations presented in [13]. As illustrated in Figure 5, we can easily gain accuracy from increased depth by leveraging MIRL for pre-training. In particular, ViT-S-54, which has the same level of computational complexity as ViT-B but is $4\times$ deeper than ViT-B, significantly outperforms ViT-B and even achieves performance on par with ViT-L pre-trained with MAE. Likewise, ViT-B-48 surpasses ViT-L while maintaining the same level of computational cost. Furthermore, the encoder pre-trained with MIRL consistently delivers higher performance than the one pre-trained with MAE.

## 4.2   Ablation studies

**Various objective functions.** Our approach is a general masking modeling architecture, seamlessly complementing prior methods that propose various objective functions. We study the compatibility of the recent feature-level loss and perceptual loss (*i.e.* VGG loss) with our method. The results listed in Table 2a show that incorporating these objective functions can further improve the model, which demonstrates the generality of MIRL. Nonetheless, the additional loss terms will introduce heavy computational overhead. To accelerate our experiments, we have not used them by default.

**Number of segments.** The purpose of dividing an encoder into more segments is to construct auxiliary reconstruction losses to facilitate the training of the intermediate layers. We observe that these auxiliary reconstruction branches can amplify the gradient magnitude, potentially improving

Table 2: MIRL ablation experiments on ImageNet-1K: We report the fine-tuning (ft) accuracy(%) for all models, which are pre-trained for 300 epochs. Unless specified otherwise, the encoder is ViT-B-24.

(a) Various objective functions. The encoder is ViT-B.

| objectives | MIRL | ft | time |
|---|---|---|---|
| pixel | ✓ | 83.5 | 1.0× |
| pixel+feature | ✗ | 83.2 | 1.6× |
| pixel+feature | ✓ | 83.6 | 1.8× |
| pixel+vgg [26] | ✓ | 83.8 | 2.5× |

(b) Numbers of segments (seg.) in the autoencoder of MIRL.

| #seg. | blocks per seg. | ft |
|---|---|---|
| 1 | 24 | 83.5 |
| 2 | 12 | 84.3 |
| 4 | 6 | 84.7 |
| 6 | 4 | 84.6 |

(c) Effect of Densely Interconnected Decoding.

| model | DID | ft |
|---|---|---|
| ViT-B-24 | ✗ | 84.5 |
| ViT-B-24 | ✓ | 84.7 |
| ViT-B-48 | ✗ | 85.0 |
| ViT-B-48 | ✓ | 85.3 |

(d) MIRL vs. simple multi-decoders. Column △ reports the performance gap between MIRL and multi-decoders.

| model | depth | MIRL | △ | multi-decoders |
|---|---|---|---|---|
| ViT-B | 12 | 83.5 | 0.3 | 83.2 |
| ViT-B-24 | 24 | 84.7 | 0.6 | 84.1 |
| ViT-B-48 | 48 | **85.3** | 0.8 | 84.5 |

(e) MIRL vs. coarse and fine separation.

| method | ViT-B | ViT-B-24 |
|---|---|---|
| MIRL | **83.5** | **84.7** |
| coarse-to-fine | 83.1 | 84.2 |
| fine-to-coarse | 82.9 | 84.2 |

Table 3: Image classification results on ImageNet-1K. All models are pre-trained and fine-tuned with 224×224 input resolution. "IN" refer to ImageNet, while "FT" is the fine-tuning accuracy. "Epochs" refer to the number of pre-training epochs. The models pre-trained with extra data or very long schedules are marked in gray. We report the best result in **bold** and the second best result(s) underlined.

| Encoder | #params | FLOPs | Method | Training Data | Epochs | FT (%) |
|---|---|---|---|---|---|---|
| ViT-B | 86M | 16.8G | Supervised | IN1K | - | 82.3 |
| | | | MoCov3 [5] | IN1K | 300 | 83.2 |
| | | | BEiT [1] | DALLE250M+IN1K | 800 | 83.2 |
| | | | SimMIM [56] | IN1K | 800 | 83.8 |
| | | | CIM [16] | DALLE250M+IN1K | 300 | 83.3 |
| | | | LocalMIM [49] | IN1K | 1600 | 84.0 |
| | | | MAE [21] | IN1K | 300/800 | 82.6/83.1 |
| | | | MAE [21] | IN1K | 1600 | 83.6 |
| | | | MIRL | IN1K | 300/800 | 83.5/84.1 |
| ViT-S-54 | 96M | 18.8G | MIRL | IN1K | 300/800 | 84.4/84.8 |
| ViT-B-24 | 171M | 33.5G | MIRL | IN1K | 300 | 84.7 |
| ViT-L | 307M | 59.7G | Supervised | IN1K | - | 82.6 |
| | | | MaskFeat [52] | IN1K | 1600 | 85.7 |
| | | | ConMIM [57] | IN1K | 1600 | 85.5 |
| | | | HPM [48] | IN1K | 800 | 85.8 |
| | | | MAE [21] | IN1K | 1600 | 85.9 |
| | | | MAE [21] | IN1K | 300/800 | 84.5/85.4 |
| ViT-B-48 | 341M | 67.0G | MIRL | IN1K | 300/800 | 85.3/**86.2** |

the optimization of deeper Transformer blocks. Table 2b reports the performance of MIRL with respect to various numbers segments. For ViT-B-24, the configuration with 2 segments shows lower accuracy than the one with 4 segments. However, further splitting the encoder into more segments brings no more performance gain.

**Effect of densely interconnected decoding.** Considering that the auxiliary reconstruction branches from different segments adopt the same objective metric, the proposed DID establishes a feature reuse mechanism, preventing layers at various depth levels from learning similar feature representations.

Table 2c demonstrates that the MIRL models embodying the DID module yield higher fine-tuning accuracy than those without the DID. Deeper models gain more advantages from DID.

**MIRL *vs*. simple multi-decoders.** In order to demonstrate that image residual learning, powered by the shortcut connections, is the key to effective training deeper ViTs, we construct a segmented autoencoder with multiple decoders. Unlike MIRL, each decoder in the multi-decoder model independently learns to reconstruct the masked content. As shown in Table 2d, MIRL achieves substantially higher accuracy than the simple multi-decoder approach. Notably, the performance gap between MIRL and the multi-decoders widens as more Transformer blocks are stacked. When pre-training with multi-decoders, the deeper ViT seems to gain accuracy from increased depth. However, this does not imply that the multi-decoder approach addresses the degradation problem. Since replacing the weights of its deeper layers with random weights does not lead to a performance drop, the trivial improvement is attributed to the increased number of shallower layers.

**MIRL *vs*. coarse and fine separation.** As the reconstructed image residual shows some fine-grained details images, it is intriguing to know what pre-training results can be produced by replacing the reconstruction targets with the coarse and fine image components separated by using a Laplacian of Gaussian operator. We construct a segmented autoencoder with multiple decoders, referred to as "coarse-to-fine", in which the reconstruction targets of the shallower and deeper segments correspond to the coarse and fine image components, respectively. "fine-to-coarse" denotes the reversed targets compared to the "coarse-to-fine" configuration. Table 2e indicates that the segmented autoencoder with fine and coarse reconstruction targets achieves lower accuracy than MIRL, demonstrating that the main and residual components are not equivalent to the fine and coarse components.

**Training schedules.** So far, we have only trained our models using a relatively short pre-training schedule of 300 epochs. Note that deeper ViTs gain more advantages from longer pre-training schedules, compared to shallower ViTs. We extend pre-training to 1000 epochs and record fine-tuning performance for various pre-training lengths. To resume pre-training from previous checkpoints, we use a step decay learning rate scheduler, decaying the learning rate by a factor of 10 at 90% and 95% of the specified pre-training length. Figure 6 shows that ViT-B tends to plateau after 800 pre-training epochs, while ViT-S-54 keeps improving even after 1000 epochs. This implies that deeper ViTs' potential can be further unleashed by adopting a very long pre-training schedule, such as 1600 epochs.

## 4.3 Image classification on ImageNet-1K

We compare our models with previous results on ImageNet-1K. Hyperparameters are provided in Appendix A. For ViT-B, MIRL pre-trained for 300 epochs achieves 83.5% top-1 fine-tuning accuracy, comparable to MAE (83.6%) pre-trained for 1600 epochs. Our pre-training is $5.3\times$ shorter, demonstrating the high efficiency of MIRL. MIRL alleviates degradation in deeper ViTs, showing impressive generalization. In an 800-epoch pre-training scheme, the deeper encoder ViT-S-54 produces 84.8% accuracy, which is 1.7% higher than ViT-B (83.1%) pre-trained with MAE and only 0.6% lower than ViT-L (85.4%). ViT-B-48, with computational complexity similar to ViT-L but $2\times$ deeper, achieves 85.3% and 86.2% accuracy with 300 and 800-epoch pre-training schemes, outperforming the ViT-L models pre-trained by other listed methods. Furthermore, the deeper encoders can further benefit from very long pre-training schedules, as discussed in Sec. 4.2.

## 4.4 Object detection and segmentation on COCO

To evaluate the generalization capabilities of our approach, we transfer our pre-trained models to the object detection task. The experiment is conducted on MS COCO [30] on account of its wide use. Following [21], we choose Mask R-CNN [22] as the detection framework and trained with the $1\times$ schedule. For fair comparisons, we adopt the identical training configurations from `mmdetection` [3], and Average Precision (AP) is used as the evaluation metric. As summarized in Table 4, MIRL outperforms all the listed methods.

## 4.5 Semantic segmentation on ADE20K

We compare our method with previous results on the ADE20K [61] dataset, utilizing the UperNet framework for our experiments, based on the implementation provided by [1] (see Appendix A for training details). The evaluation metric is the mean Intersection over Union (mIoU) averaged across all semantic categories. We employ pre-trained ViT-B-48 as the backbone, which has a computational

Table 4: Object detection results with Mask R-CNN on MS-COCO. The models pre-trained with extra data or very long schedules are marked in gray.

| Method | Backbone | Pre-training Data | Epochs | Detection $AP^b$ | Segmentation $AP^m$ |
|--------|----------|-------------------|--------|-----------------|---------------------|
| DeiT [45] | ViT-B | - | - | 46.9 | 41.5 |
| BEiT [1] | ViT-B | IN1K+DALLE | 800 | 46.3 | 41.1 |
| MAE [21] | ViT-B | IN1K | 800 | 46.8 | 41.9 |
| MAE [21] | ViT-B | IN1K | 1600 | 48.4 | 42.6 |
| MIRL | ViT-B | IN1K | 800 | 49.3 | 43.7 |
| MAE [21] | ViT-L | IN1K | 1600 | 53.3 | **47.2** |
| MIRL | ViT-B-48 | IN1K | 800 | **53.4** | 46.5 |

Table 5: Semantic segmentation results on ADE20K. The models pre-trained with extra data or very long schedules are marked in gray.

| Method | Pre-training Data | Backbone | Epochs | mIoU |
|--------|-------------------|----------|--------|------|
| MoCo v3 [5] | IN1K | ViT-L | 300 | 49.1 |
| BEiT [1] | IN1K+DALLE | ViT-L | 800 | 53.3 |
| MAE [21] | IN1K | ViT-L | 1600 | **53.6** |
| MIRL | IN1K | ViT-B-48 | 800 | 53.2 |

cost similar to ViT-L. The results are summarized in Table 5. The segmentation model using ViT-B-48 achieves competitive results compared to ViT-L pre-trained with BEiT [1] and MAE [21]. This indicates that the instantiated deeper ViTs exhibit strong transferability to downstream vision tasks.

### 4.6 Limitation and discussion

While MIRL significantly alleviates training challenges for deeper ViTs, a comprehensive theoretical explanation for the effectiveness of image residual reconstruction in training deeper ViTs remains elusive. We provide some insights into why MIRL might work well for deeper ViTs: 1) By reformulating the pre-training objective to recover the masked image's residual, MIRL implicitly encourages the model to focus on learning high-level contextual information and fine-grained details that are otherwise difficult to capture. 2) MIRL could stabilize gradient flow and enhance learning dynamics for deeper layers in ViTs, as evidenced by a larger gradient norm of the encoder in MIRL compared to vanilla MIM (see gradient visualization in Appendix C). Despite these insights, further theoretical analysis and investigation are required to fully understand MIRL's effectiveness in training deeper ViTs. The deepest ViT presented in this research comprises only 54 blocks. We anticipate that a depth of 54 is far from the upper limit for scaling ViT along the depth dimension. These areas are left for future work.

## 5 Related Work

**Self-supervised learning.** After the triumph of BERT [10] and GPT [35] models in NLP, self-supervised learning (SSL) has undergone a paradigm shift, replacing the conventional supervised learning approach [23, 43], and has made remarkable advancements in numerous domains [32, 60, 27]. Larger datasets, new training methods and scalable architectures [31, 13, 34, 59] have accelerated this growth. In computer vision, inspired by the success of BEiT [1], recent research [15, 29, 18, 24, 19, 20, 58, 2, 37, 33, 49] has explored adapting the transformer architecture to the task of image self-supervised domain. After that, the emergence of MAE [21] has further led to a resurgence of interest in reconstruction-based masked image methods, such as [21, 14, 51, 56, 11]. We are particularly intrigued by these masked methods, as they have shown state-of-the-art performance on numerous transfer tasks and are computationally efficient. This has motivated us to introduce MIRL, a novel approach that builds on these methods.

**Relation to scaling models.** Scaling deeper ConvNets [23, 43, 39, 42, 40, 41] is an effective way to attain improved performance, but the same cannot be easily achieved with ViTs [38]. While the

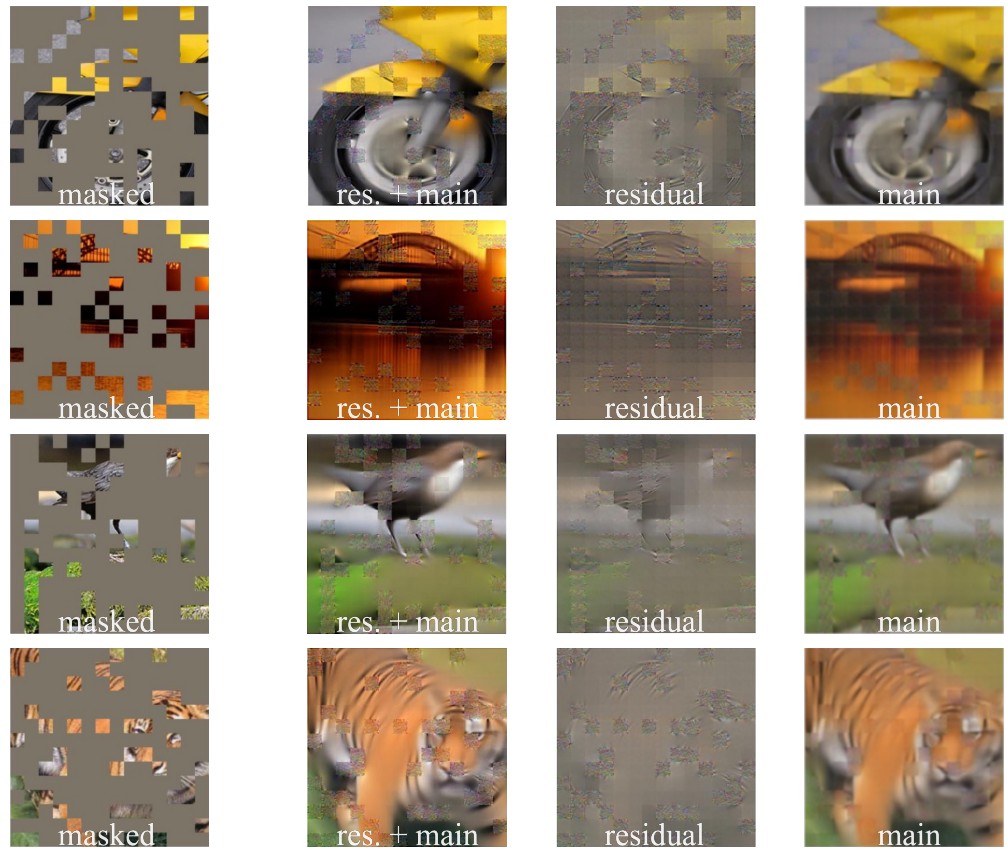

Figure 7: Visualization of MIRL. Example images are generated from the validation set on ImageNet.

Transformer architecture has succeeded in building large-scale language models [7, 53, 36, 44, 8, 17], the implementation of scalable Transformers for visual models still significantly lags behind. Recent work [62, 28, 47, 9, 59] has endeavored to explore deep Transformer-like models. These studies introduce necessary modifications to the original Transformer architecture, such as parallel layers, altered layer norm positions, composite attention mechanisms, larger embedding dimensions, unique optimization strategies, and exhaustive hyperparameter searches. Although they have demonstrated commendable performance, they lack a guiding principle about how to deepen or enlarge the Transformer-like models. Contrary to previous methods, our approach is rooted in an in-depth analysis, dissecting the standard ViT architecture. This allows us to identify the challenges in fully realizing the potential deeper ViTs and develop effective solutions accordingly. Building upon the principles we proposed, we efficiently construct deep-scale ViT models.

## 6   Conclusion

In this paper, we first reveal a performance degradation problem in Vison Transformers (ViTs) when pre-training with masked image modeling (MIM). Through an in-depth experimental analysis, we determine that the degradation is caused by the negative optimization effect of MIM enforced on deeper layers of ViT. We then introduce a novel concept of masked image residual learning (MIRL) to establish a self-supervised learning framework, aimed at alleviating the performance degradation problem. Leveraging MIRL, we unleash the potential of deeper ViTs and instantiate deeper encoders, including ViT-S-54, ViT-B-24 and ViT-B-48. These deeper ViTs variants exhibit superior generalization performance on downstream tasks.

**Broader impacts.** The proposed approach, which predicts content from training data statistics, may reflect biases with adverse societal impacts and generate non-existent content, underlining the need for further research in this area.

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
