# Masked Image Residual Learning for Scaling Deeper Vision Transformers
# Supplementary Material

**Guoxi Huang**
Baidu Inc.
huangguoxi@baidu.com

**Hongtao Fu**
Huazhong University of Science and Technology
m202173233@hust.edu.cn

**Adrian G. Bors**[†]
University of York
adrian.bors@york.ac.uk

In this Supplementary Material, we present the following items:

- Implementation details for our experiments, include settings for pre-training, image classification, object detection and semantic segmentation tasks
- Additional information about different objective function definitions for MIRL pre-training
- Additional visualization examples
- Additional information about the inference speed of deeper ViTs

## A   Implementation details

### A.1   Details for ImageNet experiments

**Pre-training.** We mostly adopt the pre-training setting in [10], except for that we adopt shorter fewer training epochs. The default pre-training setting is provided in Table 1. The learning rate $lr{=}base\_lr{\times}$batchsize / 256.

Table 1: Pre-training setting.

| Pre-training Config. | Value |
|---|---|
| optimizer | AdamW |
| base learning rate | 1.5e-4 |
| weight decay | 0.05 |
| optimizer momentum | $\beta_1, \beta_2{=}0.9, 0.95$ |
| batch size | 4096 |
| learning rate schedule | cosine decay |
| warmup epochs | 20/40 |
| training epochs | 300/800 |
| augmentation | RandomResizedCrop |

**Fine-tuning.** The layer-wise learning rate decay [4] factors of the deeper ViTs are set to larger than those of the shallower ViTs to maintain similar learning rates in the lowest layers for all models. The fine-tuning setting is provided in Table 2. In our experiment, models are insensitive to the droppath [11] configuration, and setting it to 0 does not lead to any noticeable differences. We employ EMA to enhance tuning performance on small datasets consisting of only a few hundred training samples, such as private industrial datasets. When tuning on these limited-scale datasets, we gravitate towards loading weights from the ImageNet fine-tuned model instead of the MIM pre-trained model because the MIM lacks semantic features. In this context, using the EMA-fine-tuned models yields better tuning accuracy on tiny datasets, especially when picking a non-final checkpoint, compared to

---

[†]Corresponding author. Work done when H. Fu was an intern at Baidu.

counterparts without EMA. However, EMA does not significantly impact the fine-tuning accuracy on ImageNet. The reason could be attributed to the sufficiently long training period (e.g., spanning 800 pre-training epochs + 200/100 fine-tuning epochs). This allows models with or without EMA to likely converge to the same optimum. Nonetheless, EMA remains crucial for training models from scratch, as indicated in [1] (e.g., ViT-B achieves 82.3% accuracy with EMA and 82.1% without EMA).

Table 2: Fine-tuning setting.

| Fine-tuning Config. | Value |
| --- | --- |
| optimizer | AdamW |
| base learning rate | 7.5e-4 |
| weight decay | 0.05 |
| optimizer momentum | $\beta_1, \beta_2$=0.9, 0.999 |
| layer-wise lr decay [4] | 0.65(S), 0.88(S-54), 0.65(B), 0.65(B-24), 0.88(B-48) |
| batch size | 2048 |
| learning rate schedule | cosine decay |
| warmup epochs | 20 |
| training epochs | 200(S), 100(S-54), 100(B), 100(B-24), 50(B-48) |
| augmentation | RandAug (9, 0.5) [5] |
| label smoothing [13] | 0.1 |
| mixup [16] | 0.8 |
| cutmix [15] | 1.0 |
| drop path [11] | 0.1(S), 0.1(S-54) 0.1(B), 0.1(B-24) 0.2 (B-48) |
| exp. moving average | 0.9998 |

## A.2 Details for transfer learning experiments

**Object detection on COCO.** For a fair comparison, we conduct experiments using the Mask R-CNN framework. We utilize multi-scale training and resize the image with the size of the short side between $480$ and $800$ and the long side no larger than $1333$. we initialize the backbone with the pre-trained ViT model. During fine-tuning, the batch size is $16$ and the learning rate is 1e-4. For ViT-B, the layer decay rate is $0.75$, and the drop path rate is $0.1$. For ViT-B-48, the layer decay rate is $0.88$, and the drop path rate is $0.1$. Other training configurations are adopted from `mmdetection` [2]. We do not use multi-scale testing.

**Semantic segmentation on ADE20K.** We adopt the UperNet [14] framework for semantic segmentation, following the implementation of [1]. We initialize the backbone with the pre-trained weights and fine-tune the entire model for 160k iterations with a batch size of 16. The learn rate is set to 0.0002. Different from the implementation of [1], we do not use relative position bias in our models.

# B  Other pre-training objectives

## B.1  Feature-level and VGG losses

**Feature-level loss.** Regarding the feature-level loss, we employ the InfoNCE loss used in contrastive learning:

$$\mathcal{L}^{\text{feat}} = -\log \frac{\exp(\hat{\mathbf{z}} \cdot \mathbf{z}^+/\tau)}{\exp(\hat{\mathbf{z}} \cdot \mathbf{z}^+/\tau) + \sum_{j=1}^{j=B-1} \exp(\hat{\mathbf{z}} \cdot \mathbf{z}^-/\tau)}, \tag{1}$$

where $\hat{\mathbf{z}}$ is the prediction, $\tau$ denotes a temperature parameter. In a batch with $B$ images, $(\hat{\mathbf{z}}, \mathbf{z}^+)$ represent a positive pair in which positive sample $\mathbf{z}^+$ is a momentum encoder's output on the same view of the image as $\hat{\mathbf{z}}$. The momentum encoder's parameters are the moving average of the encoder. $(\hat{\mathbf{z}}, \mathbf{z}^-)$ represents a negative pair where negative sample $\mathbf{z}^-$ is generated with an image different from that of $\hat{\mathbf{z}}$ in the image batch. Previous work in [3, 7] eliminates negative sample comparisons in their feature-level loss, which emphasizes the importance of positive samples, resembling a BYOL style [9], but we find that involving in negative samples can slightly improve the accuracy. The feature-level loss is only calculated at the end of the encoder, by appending two decoding blocks to predict the masked features.

Table 3: Comparison between loss $\mathcal{L}_g^{\dagger}$ and loss $\mathcal{L}_g$. The encoder is ViT-S-54. $\omega$ in $\mathcal{L}_g^{\dagger}$ is set to 1. We adopt a step-wise decay learning rate scheduler.

| loss definition | pre-training epochs | fine-tuning |
|---|---|---|
| $\mathcal{L}_g$ | 100 | 83.5 |
| $\mathcal{L}_g^{\dagger}$ | 100 | 83.5 |
| $\mathcal{L}_g$ | 300 | 84.2 |
| $\mathcal{L}_g^{\dagger}$ | 300 | 84.0 |

**VGG loss.** VGG loss is previously used in generative models [12, 8], eliminating the influence of pixel shifting for high-quality image synthesis. Specifically, for reconstruction from $g$-th prediction head, we replace the reconstructed patches in visible positions with the ground-truth image patches to ease the optimization difficulty, given $\tilde{\mathbf{x}} = \{\hat{\mathbf{x}}_g^i : i \in \mathcal{M}\}_{i=1}^N \cup \{\mathbf{x}^i : i \notin \mathcal{M}\}_{i=1}^N$. The mixed fake image $\tilde{\mathbf{x}}$ and growth-truth image $\mathbf{x}$ are forwarded to a fixed, lightweight VGG model, and the VGG loss is calculated by measuring the difference between their VGG activations from multiple layers, which is formulated as:

$$\mathcal{L}^{\text{vgg}} = \sum_{\ell \in \mathcal{S}} \frac{1}{C_\ell H_\ell W_\ell} \|f_\ell(\mathbf{x}) - f_\ell(\hat{\mathbf{x}}_g)\|_2^2, \tag{2}$$

where $f_\ell(\tilde{\mathbf{x}})$ denotes the activations of the $\ell$-th layer of the VGG network by inputting $\tilde{\mathbf{x}}$; $C_\ell H_\ell W_\ell$ represents the dimensions of the activation feature map, $S$ denotes a set of layers from which the VGG features are extracted. Concurrent work [6] also experiments with VGG loss.

### B.2   An alternative definition of loss $\mathcal{L}_g$

One of our early attempts regarding the form of reconstruction loss is defined as:

$$\begin{aligned}
\mathcal{L}_g^{\dagger} &= \frac{1}{|\mathcal{M}|} \sum_{i \in \mathcal{M}} \frac{1}{2P^2 C} \big( \omega \|\xi_g^i\|_2^2 + \|\xi_g^i - \hat{\xi}_g^i\|_2^2 \big) \\
&= \frac{1}{|\mathcal{M}|} \sum_{i \in \mathcal{M}} \frac{1}{P^2 C} \big( \omega \|\mathbf{x}^i - \hat{\mathbf{x}}_g^i\|_2^2 + \|\mathbf{x}^i - \hat{\mathbf{x}}_g^i - \hat{\xi}_g^i\|_2^2 \big) \\
&= \frac{1}{|\mathcal{M}|} \sum_{i \in \mathcal{M}} \frac{1}{P^2 C} \omega \|\mathbf{x}^i - \hat{\mathbf{x}}_g^i\|_2^2 + \mathcal{L}_g,
\end{aligned} \tag{3}$$

where $\mathcal{L}_g$ is reconstruction loss defined in Eq. (6) from the main paper, $\omega$ refers to the regularization weight to loss term $\|\xi_g^i\|_2^2$. This variant $\mathcal{L}_g^{\dagger}$ minimize the similarity between the original image $\mathbf{x}$ and the reconstructed image $\hat{\mathbf{x}}_g$ with reference to the first loss term, $\|\mathbf{x}^i - \hat{\mathbf{x}}_g^i\|_2^2$. By minimizing the second loss term, $\|\mathbf{x}^i - \hat{\mathbf{x}}_g^i - \hat{\xi}_g^i\|_2^2$, the deeper segment explicitly learn the image residual. In Table 3, we compare the results generated by using loss $\mathcal{L}_g$ and loss $\mathcal{L}_g^{\dagger}$. When setting $\omega$ from Eq. (3) to 1, we observe that the two different loss definitions generate similar results in a shorter pre-training period (*e.g.*, 100 epochs). However, when we pre-train the model for a longer period (*e.g.*, 300 epochs), training with loss $\mathcal{L}_g$ defined in Eq. (6) from the main paper can provide better results than loss $\mathcal{L}_g^{\dagger}$. We give the reason that $\mathcal{L}_g^{\dagger}$ determinedly minimize the distance between $\mathbf{x}$ and $\hat{\mathbf{x}}$, which could result in a very small residual. As the residual is the optimization target for those deeper segments, such a small residual due to the $\|\mathbf{x}^i - \hat{\mathbf{x}}_g^i\|_2^2$ term could corrupt the training in the deeper segments. Alternatively, by setting $\omega$ to a smaller value (*e.g.*, 0.1), we achieve a smoother optimization experience; nonetheless, the results are similar to those obtained when optimizing $\mathcal{L}_g$ alone.

## C   More Visualization

In Figure 2, we provide gradient norm visualization for MIRL and MAE. We observe that when employing MIRL for pre-training, the gradient magnitudes of Transformer blocks are larger than those when using MAE. This suggests that MIRL provides a more stable gradient flow that benefits the model optimization.

In Figure 1, we provide more visualization about image reconstruction.

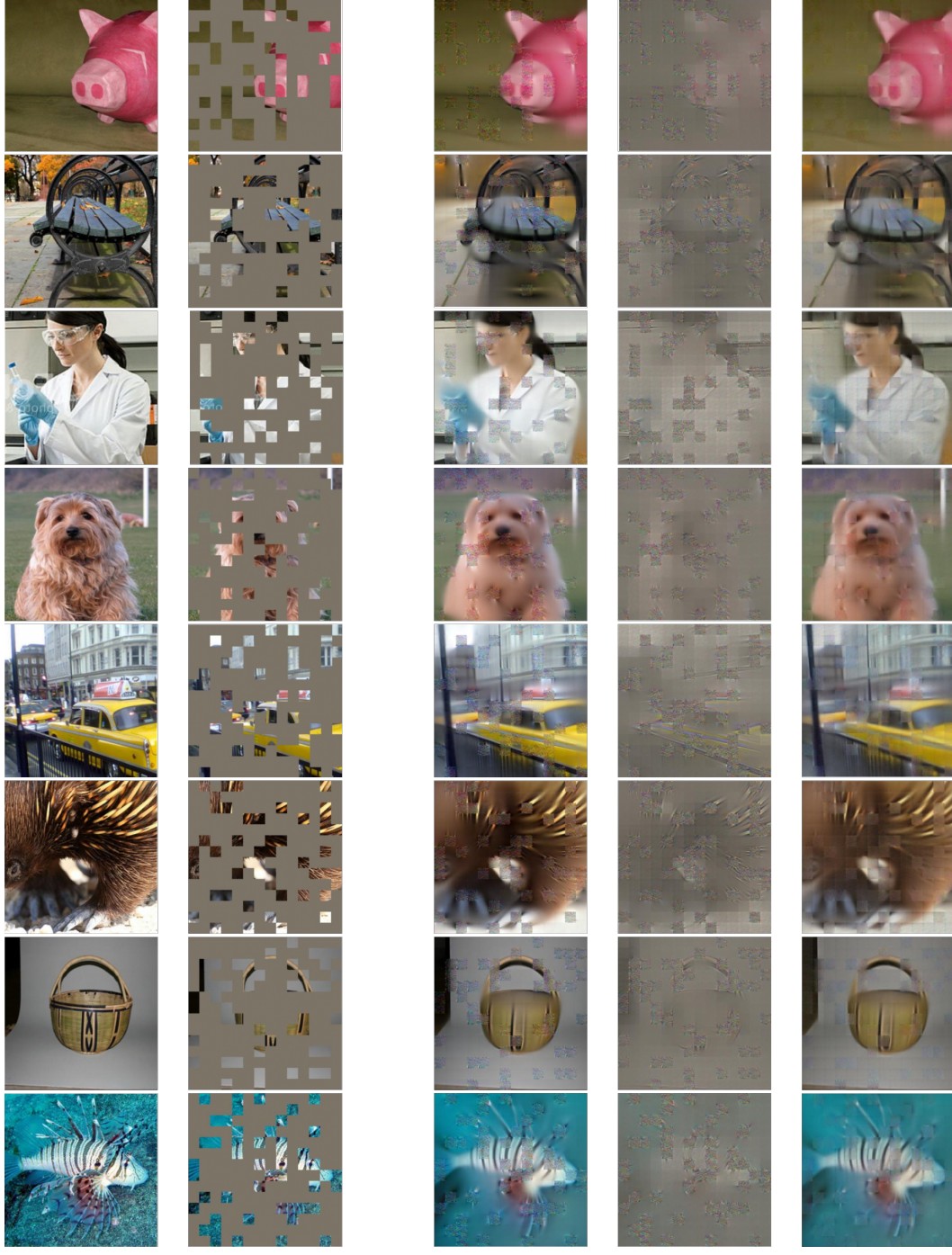

Figure 1: Example results on ImageNet validation images. For each quintuplet, we show the ground-truth, masked image, reconstruction, residual and the main component.

## D   Inference Speed

Although we have shown deeper ViTs can gain accuracy from stacking more Transformer blocks, we also notice that deeper ViTs provide lower inference speed due to the series connections between blocks. The inference speed measurement is provided in Table 4.

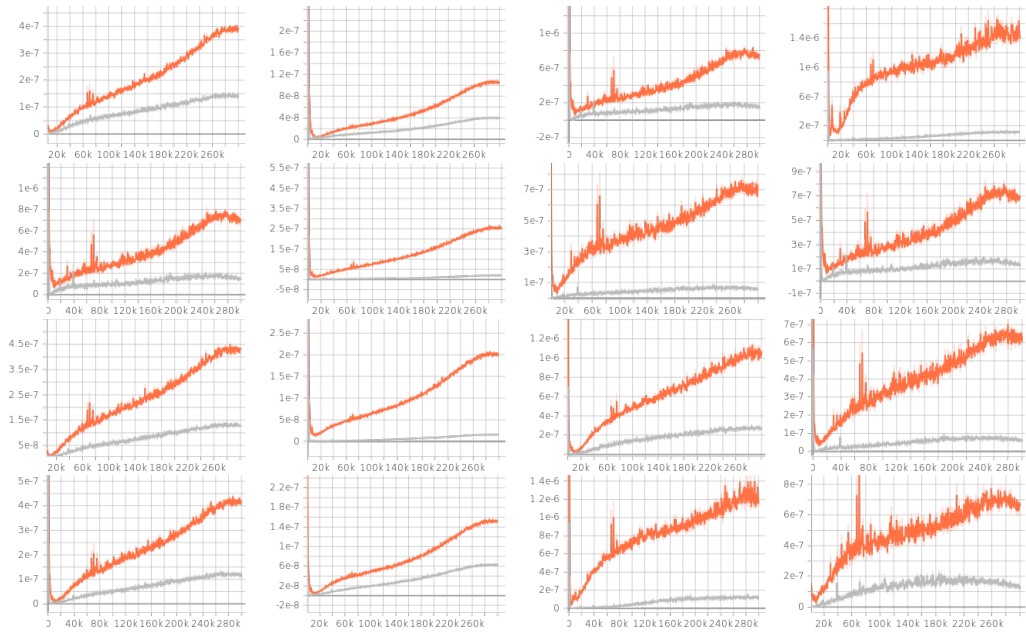

Figure 2: Example visualization of gradient norm. Original is MIRL, while gray is MAE. From top to bottom, we visualize the gradient norms in 19-22 Transformer blocks in ViT-B-24. Each row contains the gradients of attention qkv, mlp, layer norm and fc weights.

Table 4: Inference speed, measuring the throughput (images/sec) on a single V100 GPU, where the batch size is set to 256.

| model | depth | #params | FLOPs | throughput (imgs/s) |
|---|---|---|---|---|
| ViT-S | 12 | 22M | 4.2G | 751 |
| ViT-S-54 | 54 | 96M | 18.8G | 257 |
| ViT-B | 12 | 86M | 16.8G | 488 |
| ViT-B-24 | 24 | 171M | 33.5G | 285 |
| ViT-B-48 | 48 | 341M | 67.0G | 160 |

Table 5: Comparison between MIRL and truncated MIRL. For both truncated MIRL and truncated MAE, 3 blocks are not involved in pre-training, and the 5th block solely focuses on recovering the masked content.

| encoder | method | ft accuracy (%) |
|---|---|---|
| ViT-S | MIRL | 82.3 |
| ViT-S | truncated MIRL | 82.0 |
| ViT-S | MAE | 81.0 |
| ViT-S | truncated MAE | 81.7 |

## E    Whether the phenomenon in observation II still exists in MIRL?

We devise an additional model named "truncated MIRL". The concept is akin to the truncated MAE depicted in Figure 1(b) from the main paper. It involves pre-training the early encoding blocks using MIRL, while the subsequent blocks are randomly initialized. As detailed in Table 5, MIRL outperforms truncated MIRL by 0.3%. This demonstrates that MIRL effectively pre-trains the deeper layers, outperforming random initialization. This also suggests that the phenomenon observed in Observation II does not exist in the MIRL method.