# OpenReview forum: "Masked Image Residual Learning for Scaling Deeper Vision Transformers"
_NeurIPS.cc/2023/Conference — NeurIPS 2023 poster_

### Official Review · Reviewer_DfMD · 2023-06-26

**Soundness:** 2 fair
**Presentation:** 3 good
**Contribution:** 3 good
**Rating:** 6
**Confidence:** 5

**Summary:**

This paper finds that the deep layers of ViT fail to benefit from MIM pre-training. The authors replace deeper layers of MAE pre-trained ViTs with random initialization and demonstrate that this modified model achieves better performances than the original MAE pre-trained model. To this end, the authors conclude that MIM pre-training is only effective for shallow layers. To address this, Masked Image Residual Learning (MIRL) is proposed. Experiments under various benchmarks demonstrate the efficacy of the proposed MIRL method.

**Strengths:**

1. This paper is well-written and easy to follow.
2. The authors have systematically analyzed whether MIM pre-training is effective for deeper layers of ViTs.
3. The proposed method and the motivation are highly connected.
4. Experiments are sufficient and the code is provided in the supplementary material.

**Weaknesses:**

**Major Concerns:**

1. **Unfair comparisons against other methods.** I have carefully checked the fine-tuning configuration of both MIRL (Table 2 in the supplementary material) and MAE [21], and find that *MIRL incorporates an extra exponential moving average (EMA)* while others do not, which appears to be an unfair comparison. It is better to fine-tune all models pre-trained by MIRL *without* EMA. Also, several hyper-parameters have been carefully tuned compared with MAE [21], such as base learning rate, batch size, and warmup epochs. Therefore, it is better to compare with MAE which is fine-tuned under the exact same configuration.
2. **Lack of intuitive explanation of residual learning.** I recognize the efficacy of learning residual in MIM, but the underlying motivation remains unclear. In other words, given the observation that deep layers of ViT fail to benefit from MIM pre-training, why learning residual can alleviate this problem?
3. **Observation I and II cannot be reflected in Figure 2 by using a single plot named "truncated MAE".** If my understanding is correct, models of these two observations are pre-trained in different ways. Specifically, the model of Observation I should be pre-trained by vanilla MAE while the model of Observation II should be pre-trained by truncated MAE. Therefore, these two observations cannot be reflected in Figure 2. It is better to plot three curves in Figure 2, including vanilla MAE, vanilla MAE with truncated fine-tuning (Observation I), and truncated MAE (Observation II).

**Minor Questions/Suggestions:**

4. Typo in Eq. (6). It should be $||\mathbf{x}^i - \hat{\mathbf{x}}_g^i - \hat{\xi}_g^i||_2^2$.
5. It is better to combine Table 1 in the main submission with Table 4 in the supplementary material to better compare the configuration (including #params and FLOPs) of different backbones.
6. It is encouraged to plot vanilla MAE with ViT-B and ViT-L in Figure 6 to demonstrate the scaling behavior of the proposed method.
7. It is better to compare deep ViTs pre-trained with vanilla MAE.
8. The configuration of Table 2a is not clear. Is residual learning performed on both pixel and feature?
9. The setting of Table 2d is confusing. Its caption says "multi-encoder" while in lines 203-212 is "multi-decoder".


**Questions:**

I do not have any questions. Please refer to the weaknesses section. I am willing to raise my rating if the authors succeed in demonstrating that their improvements come from the proposed method and not from carefully tuned hyperparameters during fine-tuning.

**Post Rebuttal Comments**

Thanks for the rebuttal and additional comments. My concerns are well addressed. Since using EMA in fine-tuning will not affect the performance significantly, the reviewer thinks it might be a fair comparison. Also, the explanation of why residual learning works is intuitive. Therefore, I will raise my rating to 6 and the authors are supposed to revise their paper according to our dialog, including (1) clarification about the EMA, (2) an intuitive explanation about why it works, and (3) comparison with CVPR'23 works.

**Limitations:**

The authors have discussed the limitations.

---

> ### Author Rebuttal · Authors · 2023-08-09
>
> We thank the reviewer for the positive comments. We answer the questions as follows.
>
> **Q1. Unfair comparisons against other methods.  MIRL incorporates an extra exponential moving average (EMA) while others do not, which appears to be an unfair comparison ... .**
>
>
> **A1.** We address this concern from three aspects:
>
> 1. In our experiments, EMA does not significantly impact the fine-tuning accuracy on ImageNet. When fine-tuning ViTs without EMA, using the same hyperparameters as mentioned in [1], we observed similar accuracy as those obtained with EMA. For instance, when pre-trained for 300/800 epochs, ViT-S-54 achieved accuracies of 84.41%/84.79% with EMA and 84.34%/84.86% without EMA. Incorporating EMA requires adaptive adjustments to the learning rate and warm-up epochs to avoid overfitting.
>
> 2. The reason we employ EMA is to enhance tuning performance on small datasets consisting of only a few hundred training samples, such as private industrial datasets. When tuning on these limited-scale datasets, we gravitate towards loading weights from the ImageNet fine-tuned model instead of the MIM pre-trained model because the MIM lacks semantic features. In this context, using the EMA-fine-tuned models yields better tuning accuracy on tiny datasets, especially when picking a non-final checkpoint, compared to counterparts without EMA.
>
> 3. The reason EMA doesn't significantly impact performance could be attributed to the sufficiently long training period (e.g., spanning 800 pre-training epochs + 200/100 fine-tuning epochs). This allows models with or without EMA to likely converge to the same optimum. Nonetheless, EMA remains crucial for training models from scratch, as indicated in [1] (e.g., ViT-B achieves 82.3% accuracy with EMA and 82.1% without EMA).
>
> **Q2. Lack of intuitive explanation of residual learning.  Why learning residual can alleviate this problem?**
>
> **A2.** We explain the intuition behind our method as follows:
> Upon observing that deeper layers pre-trained by MIM underperformed compared to those with random initialization, we inferred that the weight parameters of these deeper layers had indeed been updated during MIM pre-training, but in an unfavorable direction.
> In contrast, the shallower layers demonstrated improved performance after MIM pre-training.  This led us to speculate that the depth of the layers could be the root cause of the degradation problem. By introducing shortcut connections between the shallower and deeper layers, we allowed the back-propagation to affect both the deeper and shallower layers simultaneously, cooperating to restore the masked content.  This MIRL design tightly couples the deeper layers with the shallower ones, implying that MIRL should either guide both the deeper and shallower layers towards a beneficial direction or lead both astray. The results suggest that the shallower layers have, in essence, steered the deeper layers towards a more favorable direction. We will include this explanation in the revised version of the manuscript.
>
> **Q3. Observation I and II cannot be reflected in Figure 2 by using a single plot named "truncated MAE". If my understanding is correct, models of these two observations are pre-trained in different ways. Specifically, the model of Observation I should be pre-trained by vanilla MAE while the model of Observation II should be pre-trained by truncated MAE. Therefore, these two observations cannot be reflected in Figure 2. It is better to plot three curves in Figure 2 ... .**
>
> **A3.** We apologize for the confusion caused. In Figure 2, we included the results for vanilla MAE, vanilla MAE with truncated fine-tuning, and truncated MAE. Within the curve for vanilla MAE in Figure 2, the initial point indicates the result of an MAE model that has been pre-trained and subsequently fine-tuned without any random initialization. This specific point aligns with the fine-tuning result of vanilla MAE.  Meanwhile, the rest points in the vanilla MAE curve indicate the results of vanilla MAE with truncated fine-tuning. We will provide further clarity on this point in the revised version.
>
> **Q4. Typo in Eq. (6). It should be $\| | \mathbf{x}^i - \hat{\mathbf{x}}^i_g - \hat{\xi}_g^i | \|^2_2.$ The setting of Table 2d is confusing. Its caption says "multi-encoder" while in lines 203-212 is "multi-decoder"**
>
> **A4.** We appreciate the reviewer for pointing out the typos. We correct them accordingly. Specifically, in the caption of Table 2d, "multi-encoders" will be corrected to "multi-decoders".
>
> **Q5. It is better to combine Table 1 in the main submission with Table 4 in the supplementary material to better compare the configuration ... .**
>
> **A5.** Thanks for the suggestion from the reviewer. We will combine Table 1 from the main submission with Table 4 in the supplementary material in the revised version.
>
> **Q6. It is encouraged to plot vanilla MAE with ViT-B and ViT-L in Figure 6 to demonstrate the scaling behavior of the proposed method.**
>
> **A6.** We agree with the reviewer's suggestion that including curves for MAE models can highlight the scaling capabilities of our method. In the revised version, we will add the curves of vanilla MAE with ViT-B and ViT-L in Figure 6.
>
> **Q7. It is better to compare deep ViTs pre-trained with vanilla MAE.**
>
> **A7.** Reviewer Pj1d raised a similar concern. Kindly refer to our response to Reviewer Pj1d's Question 5, and the table with results, for clarification. Across various model scales, MIRL consistently outperforms MAE.
>
> **Q8. The configuration of Table 2a is not clear. Is residual learning performed on both pixel and feature?**
>
> **A8.** The feature-level loss is only calculated at the end of the encoder, which means that the "residual learning " is only performed at the pixel level, as mentioned in Appendix B.1 from the supplementary material.
>
> [1] He, Kaiming, et al. "Masked autoencoders are scalable vision learners." Proc. of the IEEE/CVF CVPR, 2022.

---

> ### Comment · Reviewer_DfMD · 2023-08-15
> **Post Rebuttal Comments from Reviewer DfMD**
>
> I appreciate the rebuttal and most of my concerns are addressed.
> However, I found that the author did not compare the proposed method with some CVPR'23 works, e.g., [A] and [B]. Moreover, [B] seems to be very similar to the proposed method. The main difference lies in the reconstruction targets. The proposed method reconstruct residuals while [B] reconstructs raw RGB pixels. It is suggested to compare some new state-of-the-art methods.
>
> [A] Haochen Wang et al. Hard patches mining for masked image modeling. In CVPR 2023.
>
> [B] Haoqing Wang et al. Masked Image Modeling with Local Multi-Scale Reconstruction. In CVPR 2023.

---

> > ### Author Response · Authors · 2023-08-16
> >
> > Thanks for the reviewer's response.
> > We are truly pleased to hear that our rebuttal has helpfully addressed some concerns of the reviewer.
> >
> > We noticed the two CVPR'23 papers[1,2] mentioned by the reviewer. Given that the CVPR 2023 conference took place after our initial submission, we did not have a chance to cite them earlier. However, we are happy to cite these papers in our revised version.
> >
> > Having read the paper [2], it is evident to us that LocalMIM [2] is fundamentally different to our own work.
> > We would like to share a couple of key differences:
> >
> > 1. Our motivation is distinct from LocalMIM's. We initiated our research by uncovering a negative optimization issue in the deeper layers of networks during MIM pre-training. To the best of our knowledge, our work is the first to expose this problem, underscoring the novelty of our paper — a sentiment echoed by Reviewer 6uey.
> > Following this, our primary focus has been on tackling this degradation challenge.
> > Conversely, LocalMIM is designed for local multi-scale reconstruction where the lower and upper layers reconstruct fine-scale and coarse-scale supervisions respectively in order to accelerate the training converge speed. It is crafted with hierarchical architectures like Swin[3] in mind, which inherently contain less low-level information in the higher layers. However, the effectiveness of LocalMIM on plain Transformer architectures appears to be limited. In essence, the issue that LocalMIM tackles may not even be present in a plain Transformer architecture, as ViTs contain no downsampling operations, ensuring that low-level information is preserved in the deeper layers.
> > To shed light on this, our ablation study titled "MIRL vs. simple multi-decoders" (see Lines 203-212) provides an insightful perspective. The ablation results summarized in the table below indicate that LocalMIM (represented as multi-decoders) does not improve accuracy when used with a plain encoder architecture
> >
> >
> > 	| model  | multi-decoders | vanilla MAE | MIRL|
> > 	| -------- |  :-------:  |  :-------: |  :-------: |
> > 	ViT-B-48 |  84.5 | 84.5 |  85.3 |
> >
> > 	Additionally, based on the results provided in [1], which are listed in the table below, LocalMIM significantly improves the Swin encoder, but yields only a 0.1% accuracy increase for the ViT encoder when compared to the baseline MAE.
> >
> > 	|method| ViT-B | Swin-B |
> > 	| -------- | :-------:  | :-------: |
> > 	 MAE / GreenMIM [4] |  82.9 | 83.2 |
> > 	 LocalMIM-Pixels [2] | 83.0 | 83.7 |
> > 	 LocalMIM-HOG [2] |83.3 | 83.8 |
> >
> > 	The core feature of our framework, distinct from LocalMIM, is the image residual learning we propose. This methodology alleviates the degradation problem. The idea is straightforward yet has proven to be a highly practical solution to the degradation problem without bells and whistles.
> >
> > 2. The other distinction is that we focus on scaling deeper Vision Transformers. By leveraging MIRL, we unleashed the potential of slender ViT variants after addressing the degradation problem. As highlighted by Reviewer igwC, this is an under-explored area of research, one that LocalMIM [2] has not ventured into.
> >
> >
> > [1] Wang, Haochen, et al. "Hard patches mining for masked image modeling." Proceedings of the IEEE/CVF Conference on Computer Vision and Pattern Recognition. 2023.
> >
> > [2] Wang, Haoqing, et al. "Masked Image Modeling with Local Multi-Scale Reconstruction." Proceedings of the IEEE/CVF Conference on Computer Vision and Pattern Recognition. 2023.
> >
> > [3] Liu, Ze, et al. "Swin transformer: Hierarchical vision transformer using shifted windows." Proceedings of the IEEE/CVF international conference on computer vision. 2021.
> >
> > [4] Huang, Lang, et al. "Green hierarchical vision transformer for masked image modeling." Advances in Neural Information Processing Systems 35 (2022): 19997-20010.

---

> > > ### Comment · Reviewer_DfMD · 2023-08-17
> > >
> > > Thanks for the comments. My concerns are well addressed. Since using EMA in fine-tuning will not affect the performance significantly, the reviewer thinks it might be a fair comparison. Also, the explanation of why residual learning works is intuitive. Therefore, I will raise my rating to 6 and the authors are supposed to revise their paper according to our dialog, including (1) clarification about the EMA, (2) an intuitive explanation about why it works, and (3) comparison with CVPR'23 works.

---

> > > > ### Author Response · Authors · 2023-08-18
> > > > **Thanks**
> > > >
> > > > Thanks for the constructive feedback and recognition of our work. We will incorporate the suggestions into our revised paper.

---

### Official Review · Reviewer_6uey · 2023-07-04

**Soundness:** 3 good
**Presentation:** 3 good
**Contribution:** 3 good
**Rating:** 5
**Confidence:** 4

**Summary:**

In this paper, the authors propose a new mask image modeling method, which is helpful for deep ViT model pretraining.

**Strengths:**

Please refer to Questions

**Weaknesses:**

Please refer to Questions

**Questions:**

### strength
1. The paper is well-written and easy to follow
2. The proposed method starts from an interesting observation and is novel.
3. The experiment results seem good.

### Weakness
1. The experiment lacks some necessary reuslts. For example, ViT-L is deeper than ViT-B and has 24 layers by default, it may benefit from the proposed method. This would be a fair comparison and further prove the effectiveness of the proposed method.
2. Following 1, there is no baseline for the deeper models except Table.2(b). So when comparing with MAE, the gain of the proposed method is not clear.
3. I'm confused about the design of ViT-S-54, why use 54 layers? if it uses 48 layers, its params and FLOPs will be more similar to ViT-B and the comparison would be more fair.
4. I also have a question about the method design, why the segments are divided into two groups to predict content and residual? Take Fig4 as an example, denote input as x, if H1 predicts x0, H2 predicts x1 = x-x0, H3 further predicts x2 = x-x0-x1 and so on, such a sequential design seem more consistent with the model architecture, will it perform better?
5. A question about detail, MAE uses the pixel-normed patch as the prediction target, which is critical for its performance. Is this also used in the proposed method and the reported MAE baselines(300/800)?
6. Some baseline methods are missed. For example, MaskFeat[1] for ViT-B, Data2Vec[2], PeCO[3], BootMAE[4] and CAE[5] for ViT-B/L.
7. A small typo in Table.2(d), should it be multi-decoders?

[1] Masked Feature Prediction for Self-Supervised Visual Pre-Training

[2]data2vec: A General Framework for Self-supervised Learning in Speech, Vision and Language

[3]PeCo: Perceptual Codebook for BERT Pre-training of Vision Transformers

[4]Bootstrapped Masked Autoencoders for Vision BERT Pretraining

[5]Context Autoencoder for Self-Supervised Representation Learning



**Limitations:**

Please refer to Questions

---

> ### Author Rebuttal · Authors · 2023-08-09
>
> We thank the reviewer for finding our observations interesting and method novel. In the following, we address the concerns of the reviewer.
>
> **Q1. The experiment lacks some necessary results. For example, ViT-L is deeper than ViT-B and has 24 layers by default, it may benefit from the proposed method. This would be a fair comparison and further prove the effectiveness of the proposed method.**
>
> **A1.**
> As stated in Lines 153-157 from the paper, training ViT-L is very unstable due to the adoption of a large hidden dimension size.
> While performing MAE pre-training with ViT-L, we encountered the issue of "NaN" values. Each time when NaN occurs, we have to restart the training from the beginning epoch, as checkpoint resumptions could not solve the problem. This instability has also been reported in [1]. These training failures resulted in substantial waste of computational resources. Due to this instability, the actual computational cost for training was 2 to 3 times higher than the ideal cost, as we had to complete the pre-training of ViT-L after encountering failure in three separate attempts while using MAE.
> Nonetheless, we assure that the experimental setup, theoretical derivations, and conclusions presented in this paper adhere to rigour and reproducibility. We hold the belief that the performance trends observed for ViT-B and ViT-S can be extended to ViT-L despite the encountered instability.
>
>
>
>
> **Q2. Following 1, there is no baseline for the deeper models except Table.2(b). So when comparing with MAE, the gain of the proposed method is not clear.**
>
> **A2.** The baselines for deeper models can be found in our response to Reviewer Pj1d's Question 5, including a table with results. According to these results, across various model scales, MIRL consistently outperforms MAE.
>
>
>
> **Q3. I'm confused about the design of ViT-S-54, why use 54 layers? if it uses 48 layers, its params and FLOPs will be more similar to ViT-B and the comparison would be more fair.**
>
> **A3.** The 54-layer ViT model features a slender structure, generally deemed challenging to train. Experimenting with such a slender model can further demonstrate our approach's ability in facilitating training deeper ViT models, aligning with the paper's theme of scaling deeper vision transformers.
> We have presented a fair comparison between MAE and MIRL in Table 3 from the paper, where both employ the same encoder, specifically ViT-B.
>
>
> **Q4. I also have a question about the method design, why the segments are divided into two groups to predict content and residual? Take Fig4 as an example, denote input as x, if H1 predicts x0, H2 predicts x1 = x-x0, H3 further predicts x2 = x-x0-x1 and so on, such a sequential design seem more consistent with the model architecture, will it perform better?**
>
> **A4.** For insights into the intuition behind our method's design, kindly refer to our response to Reviewer DfMD's Question 2.
>
> The reviewer's suggestion seems to parallel a design concept where one learns the residual of another residual. Although we haven't experimented with this specific design, we harbour reservations about such a "residual-of-the-residual learning" scenario. The residual of the residual may approach zero, potentially stalling weight updates.
>
>
> **Q5. A question about detail, MAE uses the pixel-normed patch as the prediction target, which is critical for its performance. Is this also used in the proposed method and the reported MAE baselines(300/800)?**
>
> **A5.** Yes. Our models as well as the reported MAE baselines used the normalized pixel values.
>
> **Q6. some baseline methods are missed. For example, MaskFeat for ViT-B, Data2Vec, PeCO, BootMAE and CAE for ViT-B/L.**
>
> **A6.** The result for MaskFeat[2] is included in Table 3, from page 7 of the paper. However, due to space constraints in the submission, the results of the other mentioned methods are not presented in Table 3. We will include comparisons with these methods in the revised version.
>
>
>
> **Q7. A small typo in Table.2(d), should it be multi-decoders?**
>
> **A7.** Thanks for pointing out the typo. Yes, it should be multi-decoders.
>
> [1] He, Kaiming, et al. "Masked autoencoders are scalable vision learners." Proc. of the IEEE/CVF CVPR, 2022.
>
> [2] Wei, Chen, et al. "Masked feature prediction for self-supervised visual pre-training." Proc. of the IEEE/CVF CVPR, 2022.

---

> > ### Comment · Reviewer_6uey · 2023-08-18
> >
> > The rebuttal solved my concerns partly. So I keep my score as Borderline accept.

---

> > > ### Author Response · Authors · 2023-08-18
> > >
> > > Thanks for the response. We are glad to have resolved some of the concerns raised.

---

### Official Review · Reviewer_igwC · 2023-07-07

**Soundness:** 3 good
**Presentation:** 3 good
**Contribution:** 3 good
**Rating:** 6
**Confidence:** 4

**Summary:**

This paper delves into the degradation issue encountered in the deeper layers of Vision Transformer (ViT) and proposes a self-supervised learning framework called Masked Image Residual Learning (MIRL). MIRL reformulates the learning objective to recover the residual of the masked image and makes scaling ViT along depth a promising direction for enhancing performance. Experimental results demonstrate promising improvements on various downstream tasks with increased depth, highlighting the effectiveness of the MIRL method.

**Strengths:**

1.Tackling the degradation issue in training deeper vision-Transformers (ViT) in the context of MIM pre-training is a practical problem and is also a under-explored area of research.

2.This paper presents several interesting observations and shows the MIM introduces negative optimization in the deeper layers of ViT that would hinder further pre-training performances. Thre proposed method is simple and effective for solving this problem.

3.The authors conduct extensive and comprehensive experiments that show the effectiveness and efficiency of the proposed method.

**Weaknesses:**

1.While the proposed method has shown promising results, it still lacks an in-depth explanation, e.g., choosing the training objective for encouraging deep layers to recover the image residual. The rationale behind dividing different segments and providing the proposed reconstruction terms could benefit from offering additional theoretical justifications, such as utilizing tools like Mutual Information.

2.It would be beneficial to provide further clarity on the mechanism to ensure precise reconstruction of the main component and residual of the image in the shallower and deeper parts, respectively. Given that they share the same training objective, disentangling them poses a nontrivial challenge.

3.An ablation study could be added to investigate whether assigning different regularization weights to different segment loss parts is beneficial, whether the MIRL approach incurs additional training time and other costs.


**Questions:**

Training the vision-Transformers with varying scales often requires distinct hyper-parameters. For instance, the optimal hyper-parameters for training ViT-Large and ViT-Small typically differ. It would be interesting to investigate if the proposed MIRL approach can alleviate this phenomenon when training vision Transformers with varying scales.

Overall, this paper addresses an important problem and provides some insights through observations and the proposed solutions. While it leans more towards empirical findings, it would be beneficial to provide additional theoretical evidence to support the claims.

**Limitations:**

See the above comments.

---

> ### Author Rebuttal · Authors · 2023-08-09
>
> We thank the reviewer for their appreciation of our work, considering the area addressed in our paper as an important under-explored area of research. In the following we address the concerns of the reviewer.
>
> **Q1. While the proposed method has shown promising results, it still lacks an in-depth explanation, e.g., choosing the training objective for encouraging deep layers to recover the image residual. The rationale behind dividing different segments and providing the proposed reconstruction terms could benefit from offering additional theoretical justifications, such as utilizing tools like Mutual Information.**  ...
>
> **Overall, this paper addresses an important problem and provides some insights through observations and the proposed solutions. While it leans more towards empirical findings, it would be beneficial to provide additional theoretical evidence to support the claims.**
>
> **A1.** Indeed, as the reviewer mentioned and as we ourselves noted in Section 4.6, an in-depth theoretical explanation for our method remains elusive.
> An intuitive explanation for our method can be found in our response to Reviewer DfMD's Question 2.
> Additionally, we have offered some insights in Lines 259-264 from Section 4.6, entitled "Limitation and discussion," from the paper, and have included a gradient analysis in Appendix C from the supplementary materials associated with the paper. This analysis illustrates that our method could stabilize gradient flows and enhance the learning dynamics for deeper layers in ViTs.
> We plan to provide a thorough theoretical justification in a subsequent work.
>
> We appreciate the suggestions of the reviewer. We will consider analyzing the mutual information to provide more insights of MIRL.
>
>
>
> **Q2. It would be beneficial to provide further clarity on the mechanism to ensure precise reconstruction of the main component and residual of the image in the shallower and deeper parts, respectively. Given that they share the same training objective, disentangling them poses a nontrivial challenge.**
>
> **A2.** In Line 213, the ablation study provided compares MIRL with "coarse and fine" separation, examining whether a precise reconstruction of both the main component and image residual can enhance performance. The "fine and coarse" separation resembles the main component and residual separation but with two unique training objectives. Such a precise reconstruction is not as effective as our default design.
>
> In Appendix B.2 of the supplementary material submitted with the paper, we also investigated whether explicitly imposing a precise reconstruction can be beneficial by modifying the loss definition as follows:
>
> $ \mathcal{L}_g^\dag = \frac{1}{|\mathcal{M}| }\sum _{i\in \mathcal{M}} \frac{1}{2P^2C} \big(\| \xi_g^i \|^2_2 + \| \xi_g^i - \hat{\xi}_g^i \|^2_2 \big).  $
>
> This design also falls short compared to our default design. While scaling term $\| \xi_g^i \|^2_2$ with a small value such as 0.1, we achieve the same outcome as the default design.
>
>
>
> **Q3. An ablation study could be added to investigate whether assigning different regularization weights to different segment loss parts is beneficial, whether the MIRL approach incurs additional training time and other costs.**
>
> **A3.** Following the suggestion of the reviewer, we introduce weight coefficient $\lambda_g$ into our loss definition for each segment, shown in the equation below,
>
> $ \mathcal{L}_g = \frac{1}{|\mathcal{M}| } \lambda_g	\sum _{i\in \mathcal{M}} \frac{1}{P^2C} \| \xi_g^i - \hat{\xi}_g^i \|^2_2 .$
>
> We will add an ablation study for $\lambda_g$ in the revised version.
>
> MIRL does not lead to an increase in training time. Instead, throughout the paper, we emphasize the high efficiency of our method. It is not that it only delivers superior results but it also requires shorter training times than the baselines. For detailed information on training time, please refer to our response to Reviewer Pj1d's Question 2, where we also provide a table with results.
>
> **Q4. Training the vision-Transformers with varying scales often requires distinct hyper-parameters. For instance, the optimal hyper-parameters for training ViT-Large and ViT-Small typically differ. It would be interesting to investigate if the proposed MIRL approach can alleviate this phenomenon when training vision Transformers with varying scales.**
>
>
> **A4.** We appreciate the insightful suggestions. Across all ViT variants, we've maintained consistent hyper-parameter settings, which encompass learning rate, batch size, weight decay, among others, during the pre-training phase. Tweaking these hyper-parameters has a minimal impact on the performance, according to our experiments.
> During the fine-tuning phase, we found our models to be insensitive to the change of droppath configuration, as mentioned in Appendix A.1 from the supplemental material submitted with the paper.
>
> However, as we have found empirically, using a larger layer-wise learning rate decay for deeper ViTs is currently unavoidable.
> We will incorporate more details in the supplementary material of the revised version.

---

> > ### Comment · Reviewer_igwC · 2023-08-18
> >
> > I appreciate the author's efforts in providing responses, which have addressed most of my concerns. Overall, I find this work to be of good quality and interesting, and I am willing to increase my rating.

---

> > > ### Author Response · Authors · 2023-08-18
> > > **Thanks**
> > >
> > > We appreciate the recognition of our work. Thanks again for the constructive feedback.

---

### Official Review · Reviewer_Pj1d · 2023-07-08

**Soundness:** 3 good
**Presentation:** 3 good
**Contribution:** 3 good
**Rating:** 5
**Confidence:** 5

**Summary:**

This paper clarifies that MIM pretraining can induce negative optimization in deeper layers of ViT through comparison between vanilla MAE and truncated MAE. Based on this observation, this paper proposes a MAE-based framework named Masked Image Residual Learning (MIRL) for alleviating the degradation problem and training deeper ViT. In MIRL, The objective of MIM is decoupled to recovering the main component of images from the features of shallower layers, and recovering the residual from the features of deeper layers. Ablation experiments shows that MIRL outperforms MAE and MIM baselines with multiple decoders, and help deeper ViTs (e.g., ViT-S-54 and ViT-B-48) achieve better performance compared to previous methods on ImageNet.

**Strengths:**

1. In this paper, the degradation problem in the MIM pretraining of ViT is illustrated by well-designed experiments.

2. Training deeper ViT with MIM has not been widely studied in previous works.

3. The idea of recovering the residual of images is interesting, and ablation experiments show the effectiveness of MIRL.

**Weaknesses:**

1. Although this paper has cited some previous works that provide supervision to different layers of ViT (like deepMIM[1]), there is no comparison between MIRL and these previous methods in this paper.

2. Using multiple decoders in MIRL will significantly increase training time compared with MAE. A comparison of training time between MIRL and MAE should be included.

3. On the dense prediction tasks, MIRL does not show significant improvement (even weaker performance) compared to MAE with a similar number of parameters.

4. This paper says that MIRL alleviates the degradation problem in deeper layers of ViT. However, This paper does not provide evidence on whether the phenomenon in observation II still exists in the proposed method.

5. The baseline results of pretraining deeper ViTs (like ViT-S-54 ViT-B-48) with MAE are not provided in this paper.

[1] Sucheng Ren, Fangyun Wei, Samuel Albanie, Zheng Zhang, and Han Hu. Deepmim: Deep supervision for masked image modeling. arXiv preprint arXiv:2303.08817, 2023.

**Questions:**

1. The results in Table 2(d) show the comparison between MIRL and MIM baselines with multiple decoders. Did the baseline method here use DID? If not, can "residual learning + DID" significantly outperform "multi-decoders + DID"?

2. A comparison of training time between MIRL and MAE should be included.

Other questions have been mentioned in weaknesses.

**Limitations:**

The authors have addressed the limitations of this paper.

---

> ### Author Rebuttal · Authors · 2023-08-04
>
> We thank the reviewer for considering our idea interesting. We address the reviewer's concerns as follows.
>
> **Q1. Although this paper has cited some previous works that provide supervision to different layers of ViT (like deepMIM[1]), there is no comparison between MIRL and these previous methods in this paper.**
>
> **A1.** Given that DeepMIM [1] utilizes pre-trained MAE models, drawing a direct comparison with our approach may not be meaningful. We were unable to utilize the code available on DeepMIM's GitHub page. However, the DeepMIM-MAE model bears resemblance to the simple "multi-decoder model" discussed in our ablation study, the results of which are presented in Table 2(d) on page 6 of the paper. In Lines 203-212 of the paper, we discuss the results and argue that multi-decoders are not as effective as MIRL.
>
> **Q2. Using multiple decoders in MIRL will significantly increase training time compared with MAE. A comparison of training time between MIRL and MAE should be included.**
>
> **A2.** In MIRL, each decoder is quite shallow, ensuring that there isn't a significant increase in the training time. We assessed the training time per epoch on a single V100 GPU, and the total pre-training (pt) time was determined by multiplying the epoch time by the number of training epochs. As illustrated in the table below, when achieving comparable fine-tuning (ft) accuracy, MIRL requires roughly 6 times less training time than MAE. This highlights the better efficiency of the proposed methodology.
>
>
> | method  | backbone | min/ep  | #pt epochs | total pt time (hours) | ft acc. (%) |
> | -------- |  :-------: |  :-------:  |  :-------: |   :-------: | :-------: |
> MAE  | ViT-B | 64 | 1600 | 1706  | 83.6|
> MIRL  | ViT-B | 57 | 300 | 285 |  83.5 |
>
>
> **Q3. On the dense prediction tasks, MIRL does not show significant improvement (even weaker performance) compared to MAE with a similar number of parameters.**
>
> **A3.** For the semantic segmentation task on ADE20K, MIRL yields an mIOU that is 0.4 lower than MAE's. However, our pre-training schedule is only half the duration of MAE's (i.e., 800 epochs vs. 1600 epochs). Given these closely matched results, the greater efficiency of MIRL deserves emphasis.
> Moreover, when pre-training for a longer schedule, we expect our model to deliver even better performance, as analyzed in Figure 6 from the paper. Additionally, even with fewer pre-training epochs, our method outperforms other approaches in both image classification and object detection tasks.
>
>
>
> **Q4. This paper says that MIRL alleviates the degradation problem in deeper layers of ViT. However, This paper does not provide evidence on whether the phenomenon in observation II still exists in the proposed method.**
>
>
> **A4.** To address this concern, we devise an additional model named "truncated MIRL". The concept is akin to the truncated MAE depicted in Figure 1(b) from the paper. It involves pre-training the early encoding blocks using MIRL, while the subsequent blocks are randomly initialized. As detailed in the table below, MIRL outperforms truncated MIRL (in which 3 blocks are not involved in pre-training, and the 5th block solely focuses on recovering the masked content) by 0.3%. This demonstrates that MIRL effectively pre-trains the deeper layers, outperforming random initialization. This also suggests that the phenomenon observed in Observation II does not exist in our method. Due to the space limitation for the paper, an additional diagram to depict truncated MIRL will be provided in the supplementary material of the revised version.
>
>
>
> | encoder | method  |  ft acc. (%) |
> | -------- |--------  |  :-------: |
> | ViT-S | MIRL |  82.3 |
> | ViT-S | truncated MIRL |  82.0 |
> | ViT-S | MAE |  81.0 |
> | ViT-S | truncated  MAE |  81.7 |
>
>
> **Q5. The baseline results of pretraining deeper ViTs (like ViT-S-54 ViT-B-48) with MAE are not provided in this paper.**
>
> **A5.** We have performed the relevant experiments and in the table below we provide comparisons between MAE and MIRL when using deeper ViTs.
>
>
> | encoder | method  | pre-training epochs|  ft acc. (%) |
> | -------- |--------  | :-------: | :-------: |
> | ViT-S-54 | MAE | 300 |  82.7 |
> | ViT-S-54 | MIRL | 300 |  84.4 |
> | ViT-B-48 | MAE | 300 |  84.5 |
> | ViT-B-48 | MIRL | 300 | 85.3 |
>
> By tackling the degradation problem, MIRL consistently outperforms MAE and offers superior model scaling capabilities.
>
>
> **Q6. The results in Table 2(d) show the comparison between MIRL and MIM baselines with multiple decoders. Did the baseline method here use DID? If not, can "residual learning + DID" significantly outperform "multi-decoders + DID"?**
>
> **A6.** To address this concern, we quantize our method as MIRL =  multi-decoders + DID + "residual learning".
> After accounting for each of these components, as well as their various combinations, we provide the following results:
>
> | model  | multi-decoders  | DID | residual learning |Top-1 (%) |
> | -------- |  :-------:  |  :-------: | :-------: | :-------: |
> ViT-B-48  |&cross;| &cross;| &cross;  |  84.3 |
> ViT-B-48  |&#10004;| &cross;| &cross;  |  84.5 |
> ViT-B-48  |&#10004;|&#10004;| &cross;  |  84.5 |
> ViT-B-48  |&#10004;|&cross;| &#10004;  |  85.0 |
> ViT-B-48  |&#10004;|&#10004;| &#10004;  |  85.3 |
>
> We clarify that in Table 2(d) from the paper, the multi-decoder implementation utilizes DID. However, as indicated in rows #2 and #3 of the above table, DID does not enhance the model's performance when "residual learning" is excluded. As we mentioned in Lines 116-120 from the paper, "residual learning" represents the core of the proposed framework. Still, DID boosts the "multi-decoders" accuracy by ~0.1% when the encoder is ViT-B-24.
>
>
>
> [1] Ren, Sucheng, et al. "DeepMIM: Deep Supervision for Masked Image Modeling." arXiv preprint arXiv:2303.08817 (2023).

---

> > ### Comment · Reviewer_Pj1d · 2023-08-21
> >
> > Thanks for authors’ effort in the rebuttal. The responses have addressed most of my concerns. I will keep my rate and recommend accepting this paper.

---

> > > ### Author Response · Authors · 2023-08-21
> > > **Thanks**
> > >
> > > We are glad to have resolved some of the concerns raised. Thanks for the recognition of our work.

---

### Official Review · Reviewer_Cuwd · 2023-07-10

**Soundness:** 2 fair
**Presentation:** 2 fair
**Contribution:** 2 fair
**Rating:** 5
**Confidence:** 5

**Summary:**

* This paper propose a Masked Image Modeling pretraining framework,  which boost the pretraining performance in deeper deeper layers of ViT architecture.

* Deeper vision transformers are hard to train, so authors introduce a new pretraining objective for mim, which can  alleviate the degradation problem in deeper vit.

* MIRL segment the encoding blocks into several blocks(such as 4) and use different decoders to reconstruct the target.

* various experiments are conducted, with less pre-training time, MIRL can achieve competitive performance compared to other mim approaches especially in deeper vit.

**Strengths:**

* well motivated and well written, training and pretraining in deeper vit architecture is important, experiments show that pretraining in deeper vit-base can get similar result with vit-base.

* The downstream task performance(in table 4) show superior performance than MAE framework, +2.5 bbox mAP, +1.8 mask AP.

**Weaknesses:**

* Since MIRL with deeper layers is different from the standard vit architecture(more deep than original vit), the fully supervised performance of the the same layers should be reported, the compare is a little in Table 3.

* I am interested that would this framework be also work in convnet mask image modeling framework(such as simmim, spark) ?


**Questions:**

please refer to the weakness

---

> ### Author Rebuttal · Authors · 2023-08-04
>
> **Question1 (Q1). Since MIRL with deeper layers is different from the standard vit architecture(more deep than original vit), the fully supervised performance of the the same layers should be reported, the compare is a little in Table 3.**
>
> **Answer 1 (A1).** We provide an additional comparison between ViT-B and ViT-S-54, shown in the below table, illustrating that a deeper ViT does not exceed a standard ViT when they do not experience the masked unsupervised training. We opt to not present these results in our original submission because we want readers to put more attention to the degradation problem that happened in the Masked pre-training process.
> We use the same hyperparameters as provided in [1].
> | Encoder | Method | ImageNet (%) |
> | -------- | -------- | :-------: |
> | ViT-B   | supervised    |              82.3|
> | ViT-S-54  | supervised   |         82.2|
>
> **Q2. I am interested that would this framework be also work in convnet mask image modeling framework(such as simmim, spark) ?**
>
> **A2.** Since our code was originally implemented for the ViT arch, we have not tested our MIRL framework by using pure ConvNets. However, we instead experiment with the hybrid architecture, ConvViT[2, 3], which is made up of both conv layers and transformer layers. When using ConvViT-B as an encoder, using a 300 epoch-pretraining scheme, MIRL generates 85.1\% finetuning accuracy higher than the baseline (84.3\%).
>
>
> [1] He, Kaiming, et al. "Masked autoencoders are scalable vision learners." Proceedings of the IEEE/CVF conference on computer vision and pattern recognition. 2022.
>
> [2] Gao, Peng, et al. "Convmae: Masked convolution meets masked autoencoders." arXiv preprint arXiv:2205.03892 (2022).
>
> [3] Dosovitskiy, Alexey, et al. "An image is worth 16x16 words: Transformers for image recognition at scale." arXiv preprint arXiv:2010.11929 (2020).

---

> > ### Comment · Reviewer_Cuwd · 2023-08-21
> >
> > Thank you very much for your detailed explanation of my concerns. I keep my initial score.

---

> > > ### Author Response · Authors · 2023-08-21
> > >
> > > Thanks for the response and recognition of our work.

---

### Decision · Program_Chairs · 2023-09-21

**Decision:**

Accept (poster)

**Comment:**

This paper focuses on the issue of performance degradation in the deeper layers of Vision Transformer (ViT) when applying masked image modeling. It introduces a new framework called MIRL. The paper underwent evaluation by five experts in the field, all of whom expressed strong interest in the problem being studied. Despite initial concerns raised before the rebuttal phase, the authors successfully addressed these concerns and received unanimous positive feedback from the reviewers.

The Area Chair agrees with the reviewers that the problem addressed in this paper is of great significance and has the potential to inspire further research in the field, thus recommending acceptance. The authors should seamlessly integrate all the newly added experiments and analyses into the final version of the manuscript.